# Shank Circumference Reduction by Sleep Compression Stockings in University Students and Convenience Store Cashiers

**DOI:** 10.3390/healthcare10081532

**Published:** 2022-08-13

**Authors:** Yi-Lang Chen, Pai-Sheng Huang, Che-Wei Hsu, Yuan-Teng Chang, Hong-Tam Nguyen

**Affiliations:** 1Department of Industrial Engineering and Management, Ming Chi University of Technology, New Taipei 24301, Taiwan; 2Buy2sell Vietnam, Ho Chi Minh City 70150, Vietnam

**Keywords:** cashiers, compression stockings (CSs), discomfort, shank circumference (SC)

## Abstract

Compression stockings (CSs) are a relatively simple and effective tool for alleviating varicose veins and are often used as a preventive measure among workers whose jobs require prolonged standing. Nevertheless, the efficacy of CSs that are advertised as sleepwear remains unverified. This study recruited 10 female university students and 10 cashiers as participants to test the effects of sleep CSs. During the experiment, the changes in shank circumference (SC) and the subjective discomfort rating upon getting up and going to bed were collected. Data were recorded immediately after getting up and SC measurement was repeated 10 min later. The results demonstrated that both CS condition and measurement time significantly affected SC reduction, whereas cashier or student status did not. The reported discomfort and tightness of the legs attributed to CSs were relatively high, and the benefit toward SC reduction was minimal. Cashiers exhibited slightly larger SC values and higher perceived discomfort levels, which may be attributed to their occupational characteristic of prolonged standing, and the cumulative effect of prolonged standing on muscle properties warrants further study. The study findings suggest that wearing CSs for sleep may not be effective for reducing OE

## 1. Introduction

Varicose veins (VVs) are a highly prevalent vascular disorder [1,2] that is often neglected until its serious symptoms emerge. This disorder is closely associated with prolonged standing [3]; thus, jobs requiring prolonged standing are among the risk factors affecting its development. The results of a large-scale survey in Denmark with 1.6 million participants revealed that among men whose jobs required prolonged standing, the risk ratio for VV development was 1.85, and the corresponding risk ratio for women was approximately 2 times higher than [3] that of men. An extensively cited survey by the German Federal Institute for Occupational Safety and Health revealed the high prevalence of work requiring prolonged standing in Germany. Approximately 55% of employees surveyed indicated that they frequently worked from a standing position [4]. Similar results were reported in other surveys [5].

Various reports have documented the prevalence of VVs in different populations. The estimated prevalence of VVs in adults varies widely, ranging from 7% to 40% in men and from 14% to 51% in women [6,7]. The prevalence of VVs was also reported as 18.7% in individuals of Asian ethnicity [8]. Moreover, the relationship between prolonged standing and VVs has been established in multiple studies of various at-risk professional groups, including nurses [9,10], hairdressers [11,12], teachers [13], and cashiers [14].

Occupational edema (OE) of the lower limbs has been associated with VVs and may be its precursor. Weddell [15] indicated that the skin signs such as edema and varicose ulcer are significantly more common in those with VVs than in those without. Even in venous healthy people, there was a significant increase in lower leg volume after a short (15 min) period of controlled standing [16], which may also be a potential risk of VV. This disorder is commonly observed in people whose jobs involve performing tasks requiring prolonged standing. These tasks lead to varying degrees of lower limb loading [17] and are associated with the symptoms of leg pain, swelling, heaviness, and disturbing sensations [18]. Lower limb compression has been used to treat VVs and OE for at least four decades [19]. In modern clinical settings, compression stockings (CSs) are the recommended treatment for VVs and OE [20,21]. Medical compression therapy with CSs is considered a conservative therapeutic standard in edema therapy [16]. Additionally, the use of CSs constitutes a simple and convenient home intervention that reduces OE symptoms without severe or long-lasting side effects [22]. Generally, CS therapy improves blood circulation in the legs by increasing pressure. CSs are available in a variety of styles and their main function is to compress the legs. Health-care professionals can readily prescribe a pair of appropriate CSs for the treatment or prevention of VVs [12]. On this precedent, many people who engage in tasks requiring prolonged standing have taken the initiative to purchase nonprescription CSs as a preventive measure. Some individuals have also adopted CSs as a beautification device for calf shaping.

Because OE is potentially associated with VVs, the efficacy of CS use for eliminating OE has been examined through clinical studies. Relevant studies have reported changes in the physiological measurements of the lower limbs before and after wearing CSs, demonstrated through foot volumetry and air plethysmography [4,18,23]. Another simple method of deducing the ameliorative effect of CSs is through the measurement of the reduction in shank circumference (SC), which assesses changes in the loading and swelling of the lower legs under various test conditions [24,25,26].

Workers in many occupations are vulnerable to VVs. Patients who already have VVs must wear CSs as a component of postoperative therapy, whereas those using CSs for preventative or aesthetic purposes can wear CSs at their convenience. CS use may limit movement and interfere with activities or cause distracting sensations such as overheating, irritation, skin dryness, and itchiness [27,28,29]. Consequently, those using CSs voluntarily may opt to wear them during nonworking hours only. Manufacturers have accommodated this market by introducing CSs designed for use while sleeping to relieve leg swelling and discomfort that accumulates during the day while simultaneously preventing VVs. Sleep CSs are popular among their intended audience, but their efficacy remains unclear.

Cashiering is a prime example of a job that increases vulnerability to OE and VVs. With respect to the cashier’s job characteristics, operating from a standing position is beneficial for minimizing postural stress, fatigue, and discomfort [14]. Nevertheless, the prolonged standing undoubtedly increases lower limb loading in these workers, and because cashiers often remain unmoving behind the checkstand for hours on end, their relatively low activity levels may also increase the degree of leg edema [30]. Accordingly, this study recruited 10 female university students and 10 female convenience store cashiers to examine the effects of sleep CSs and to quantify the perceived discomfort levels associated with wearing CSs and compare the differences in the efficacy of wearing CSs between the two groups. In addition to the cashiers, university students were recruited in the test because they can be considered as being representative of the non-prolonged standing group, and they are also one of the main consumers of sleep CSs for calf shaping. In this study, we hypothesized that wearing CSs, measurement time, and participant group would affect the SC reduction. The results are a reference for sleep CS use. 

## 2. Materials and Methods 

### 2.1. Participants

A total of 20 female participants with no history of musculoskeletal disorders were enrolled in this study; all participants received a daily wage as compensation for completing the experimental tasks. Ten cashiers with a minimum of 1 year of experience were selected from chain convenience stores; these cashiers worked approximately 8 h per day during the daytime shift. Ten student participants were recruited from a local university where they were actively engaged in classes and general campus activities. The mean (standard deviation (SD)) age, height, and bodyweight were 22.7 (2.5) years, 159.6 (7.3) cm, and 52.9 (5.9) kg, respectively, for the cashiers and 21.9 (1.5) years, 158.7 (4.4) cm, and 51.5 (4.1) kg, respectively, for the students. An independent *t* test revealed no differences in the basic data between the two groups, with the exception of their SC measurements (Table 1). Written informed consent was obtained from all participants in accordance with the Declaration of Helsinki. This study was approved by the Ethics Committee of National Taiwan University.

### 2.2. SC Measurement

The SC measurement was conducted using a modified version of the method discussed in the relevant literature [24,25,26]. To minimize measurement errors, a pull–push tester (MP-1; Attonic, Aichi, Japan) was used to control the constant force applied during the measurement process (Figure 1a). The measurement area was marked off close to the knee and at one-third of the length between the knee and the ankle joints. These points remained marked for the duration of the study. During the measurement process, participants were asked to stand on the ground, and each SC measurement was repeated twice. The average measurement was then used for further analyses.

### 2.3. Visual Analog Scale for Discomfort and Tightness Rating

In this study, subjective assessments of lower limb discomfort were performed using a continuous visual analog scale (VAS), with a score from 0–10. The VAS is considered a reliable assessment of perception and is more precise than an ordinal scale that ranks responses; the VAS used in the present study was modeled after the comfort scales developed by Mündermann et al. [31]. The left end of the scale was labeled “no discomfort at all” and the right end was labeled “extreme discomfort”. In addition to discomfort, participants were asked to assess the tightness caused by wearing CSs during sleep. The left and right scales were labeled “no tightness at all” and “extreme tightness”, respectively. The participants used a pen to mark the locations on the scales that most accurately represented their feelings of discomfort and tightness. A research assistant used a ruler to measure the distance from the “no discomfort (or tightness) at all” anchor to the location of the pen mark, and this distance was converted into a numeric score (0–10) for further analyses. This VAS method has been regularly employed in discomfort evaluations, including in our own recently published studies [26,32].

### 2.4. Experimental Design and Procedure

This study collected SC data and subjective discomfort and tightness scores for the lower limbs under two CS conditions (with and without wearing CS, Figure 1a,b) over multiple sessions. Data from our trials (2 CS conditions × 2 measurement times × 2 repetitions) were collected for the 10 female students and the 10 female cashiers. SC reduction was compared based on the data measured immediately after getting up and that collected 10 min later. The CS (MediQttO Sleep & Full Leg, 4-covered parts; Dr. Scholl, Tokyo, Japan) used in this study was recognized as a highly popular nonprescription CS product among cashiers. The CS was specifically designed to be worn during sleep, with pressures ranging from approximately 15 to 20 mmHg. This leg compression range was rated as effective for treating VVs [33]. Participants chose the most suitable stocking size based on their thigh and shank circumferences, in accordance with the manufacturer’s instructions, and confirmed the fit by trying on the product.

For SC measurement during each session, each participant was paired with a well-trained research assistant. The interclass reliability among the research assistants was previously established as satisfactory through a training program and pilot tests. Under the condition that no residual effect from prior CS use remained, the SC of each participant was measured immediately in the morning at the beginning of the study period and then again before going to bed. The difference between measurements was regarded as the baseline SC increment during a normal day of student or cashier activities (Table 1). Subsequently, the participants’ SC values were measured either with or without CS intervention. The 20 participants were randomly divided into two groups (groups A and B), with five students and five cashiers in each group (Figure 2). After a prescribed sleep time of 7 to 8 h, all participants were asked to perform their regular activities (cashiering or class attendance) and then return to their home or dormitory as usual. During the testing period, participants were instructed not to engage in excessively strenuous exercise. At bedtime, the participants in groups A wore sleep CSs, whereas those in group B slept without wearing CSs. Upon getting up the next day, each participant immediately recorded their SC and the VAS measurements for their lower limbs. The participants in group A (the CSs group) also completed a VAS reporting the tightness attributed to CSs during sleep. After 10 min of routine morning activity (brushing and washing, etc.), all participants once again measured their SC. After an interval of 3 days, the two groups switched, so that group B became the CS group. The measurement process was repeated twice during each 3-day session, and the average SC value of each participant was used for further analyses. All calculations of SC reduction were calculated on an individual basis using these measurements. 

### 2.5. Statistical Analysis

All data analyses were performed using SPSS Statistics (v.23.0, IBM, Armonk, NY, USA), with a significance level of α = 0.05 for all tests. Participant data were analyzed using descriptive statistics (i.e., mean, SD, and independent *t* test). A three-way analysis of variance (ANOVA) was used to examine the effects of the two CS conditions (wearing and not wearing CSs) and the two SC measurement times (getting up and 10 min after) on SC reductions and VAS values. SC reductions were obtained by subtracting the SC measured at each time point from the SC before going to bed. We also used the Pearson product–moment correlation coefficient (*r*) to examine the repeatability of each SC measurement, and we used independent and paired *t* tests to identify any differences between sessions in basic participant data, the SC values, and the discomfort (or tightness) scores.

## 3. Results

### 3.1. Baseline SC Measurements

Table 1 presents the baseline SC measurements for each group. The SC values for the cashiers were significantly higher than those of the students, regardless of when the measurement was taken (i.e., after getting up or before going to bed, both *p* < 0.05). However, the two groups exhibited nonsignificant differences in increased SC values after a typical working or school day (students, 3.0 mm; cashiers, 3.1 mm).

### 3.2. Effects of Variables on SC Reduction

The measurement repeatability of the two obtained SC values for all participants was 0.935 (*p* < 0.01), indicating satisfactory consistency. Figure 3 illustrates the SC values collected at varying time points under wearing and without CSs for the two participant groups, and the SC reductions were thus calculated. The three-way ANOVA revealed that CSs (*p* < 0.001) and the time of measurement (*p* < 0.05) significantly affected SC reduction (Table 2) and that both groups experienced a similar SC reduction (students, 4.4 mm; cashiers, 3.4 mm), when averaged across other variables. Wearing CSs resulted in an overall higher SC reduction during sleep (5.4 mm) than not wearing CSs (2.4 mm). SC was reduced to 4.8 mm after getting up; however, this reduction declined significantly to 2.9 mm after 10 min. Figure 4 further illustrates the SC reductions under various combinations of variables (cashier vs. student, wearing vs. not wearing CSs, and measurement time).

### 3.3. Discomfort and Tightness Results

Table 3 reveals the results of the VAS assessments across both sessions. The differences between the two sessions were examined using an independent or paired *t* test. The results indicated that the cashiers (score: 4.8) experienced higher levels of leg discomfort than the students (score: 2.9) before going to bed (*p* < 0.05). After getting up the next morning, no significant differences in tightness attributed to wearing CSs were recorded. However, tightness attributed to wearing CSs for both the students and the cashiers was relatively high, with scores of 4.2 and 5.1, respectively. Only the students who did not wear CSs reported lower discomfort levels after getting up (score: 2.0) than before going to bed (score: 2.9; *p* < 0.05). Additionally, the cashiers reported significantly higher levels of leg discomfort than the students did, regardless of whether they wore CSs or not (*p* < 0.05 and < 0.01, respectively).

## 4. Discussion

For treating OE, CSs are usually worn during working hours [34], and studies have revealed that CSs significantly alleviate leg OE. Some studies have also demonstrated that wearing CSs during the day inhibits nighttime leg swelling [35]. However, for healthy people whose jobs require prolonged standing and who use CSs only as a preventive measure, wearing CSs while working may interfere with movement and cause discomfort, especially in hot environments [11,36]. Therefore, sleeping in CSs was proposed as an alternative approach. However, the preliminary results of this study did not exactly match our hypotheses. The results revealed no significant difference in SC reduction in either the student group or the cashier group after sleeping in CSs. Although wearing CSs for 8 h overnight induced a mild SC reduction, the effect quickly dissipated within 10 min after getting out of bed (a 26% decline in both groups). When not wearing CSs, the reversion in SC reduction was much higher in the student group (46.6%) than in the cashier group (29.8%; Figure 4). Wearing CSs can cause tightness and discomfort, and because the benefits of reducing lower limb load and edema were minimal, wearing CSs during sleep is not recommended.

After a day of normal activities, at bedtime, the lower limb edema of the cashiers was no higher than that of the students (both approximately 3 mm). We did not measure the participants’ SC immediately after getting off work because the purpose of this study was to observe the effects of overnight CS intervention. Previous research has revealed that both standing and sitting can cause lower limb swelling [4,37]. This may explain why no significant differences in lower limb edema were observed between the student and cashier groups. Although students may have more opportunities to move around during the day while walking from class to class, Garcia et al. [38] suggested that simply altering the work–rest cycle may not be sufficient to counteract the lower limb loading; only continuous exercise for at least 30 min can offset the cumulative effect of postural edema [21]. Table 1 reveals that the overall SC values of cashiers were significantly higher than those of the students, with differences of more than 31 mm. No significant differences in body sizes were noted between the two groups, with the exception of the SC values. This finding may be attributed to the occupational characteristics of the cashiers. The cashiers in this study had at least 1 year of work experience, and the cumulative effect of prolonged standing may have resulted in a larger SC value; however, no study evidences this theory. Prolonged standing induces marked changes in the lower leg muscle twitch force, force control, muscle oxygenation, postural stability, perceived discomfort, and volume measurements immediately after 5 h of work, implying a detrimental effect in long-lasting muscle fatigue, performance, discomfort, and vascular aspects [38]. Chen et al. [25] discovered that participants with larger SC values had approximately a 2 mm less SC increase after 4 h of static standing relative to participants with normal SC values. They theorized that the long-term accumulation of limb loading in the participants with larger SC values may have affected the musculature characteristics of the lower limbs, thus differentiating them from the participants with smaller SC values. The present study also found that the effect of wearing CSs on SC reduction for the cashier group was not as notable as it was for the student group (students: 7.9 mm, cashiers: 5.2 mm; Figure 4); however, most of the cashiers perceived more severe discomfort in their legs than the students during the study period, regardless of whether they wore CSs (Table 3). People working jobs involving prolonged standing may feel leg pain, bloat, heaviness, and other distracting sensations [18]. The changes in muscle properties due to specific occupational characteristics warrant further investigation.

Although CSs are primarily used to reduce OE of the lower limbs, wearing CSs while sleeping is generally believed to shape the curves of the legs. Although this study could not completely disprove the existence of a leg-shaping effect, the effect of wearing CSs on SC reduction was less pronounced than expected, suggesting that the influence of CSs on the beautification of leg curves would also be minimal. Overall, wearing CSs while sleeping did not ameliorate participants’ subjective feelings of discomfort, especially in the cashier group. Although relatively low-pressure CSs were adopted for this study, previous studies have demonstrated that low-pressure CSs were sufficient to provoke efficacy [34], and higher-pressure CSs likely would not have demonstrated any additional benefits for our participants [23,33].

This study has several limitations. First, the relatively small sample size (*N* = 20) may limit the generalizability of our findings. Even though the effect sizes (power in Table 2) for the analyses were close to or more than the criteria (i.e., 0.8) suggested by Cohen [39], this small sample definitely alters the robustness of the results. Second, although various CS sizes were available for the participants to choose from, not all participants found CSs that fit their leg shapes, and this may have reduced the product efficacy. The CSs used in this study had a pressure ranging from approximately 15 to 20 mmHg. The effect of sleep CSs on the responses might be different while using higher or lower pressure CSs. Third, many methods for evaluating leg loads and quantifying OE exist. This preliminary study employed SC measurement because of its simplicity and low cost, but the validity of this method compared with other protocols requires future examination. Fourth, although the participants were asked to maintain a strict schedule and avoid intense exercise, it was impossible to completely control for variations in the activities of the participants throughout the day. Finally, our data are insufficient to draw conclusions regarding the efficacy of sleep CSs for leg shaping.

## 5. Conclusions

Unlike previous studies that focused on wearing CSs during the day to directly inhibit OE or reduce nighttime swelling, this study examined the influence of wearing nonprescription sleep CSs on SC reduction in two groups (students and cashiers) and the accompanying subjective perception of discomfort. The preliminary results demonstrated that although wearing CSs had a significant effect on SC reduction, the effect rapidly declined by more than one-quarter within 10 min after getting out of bed. Moreover, wearing CSs may cause discomfort and tightness during sleep. Thus, the findings based on the SC reduction indicate that wearing CSs for sleep is not effective for reducing OE, and we do not recommend the use of sleep CSs.

## Figures and Tables

**Figure 1 healthcare-10-01532-f001:**
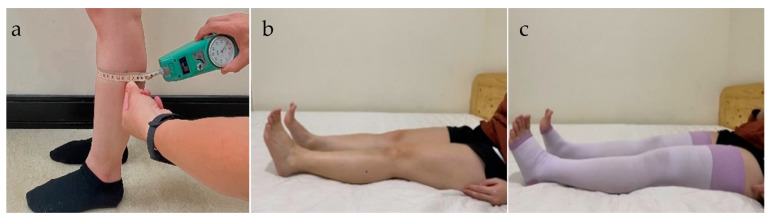
Schematic of measurement for shank circumference (**a**), and participants not wearing (**b**) and wearing compression stockings (**c**).

**Figure 2 healthcare-10-01532-f002:**
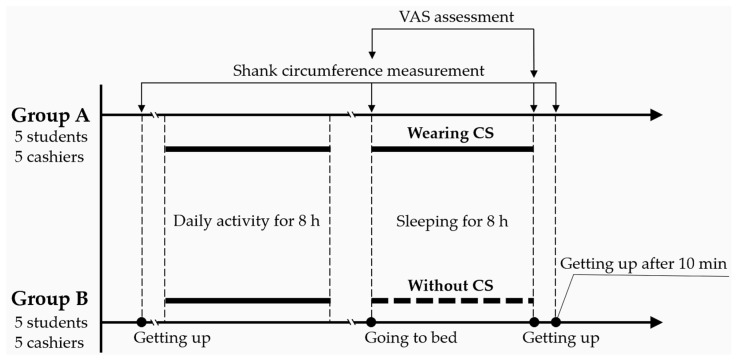
Schematic timeline of the study design (VAS, visual analog scale).

**Figure 3 healthcare-10-01532-f003:**
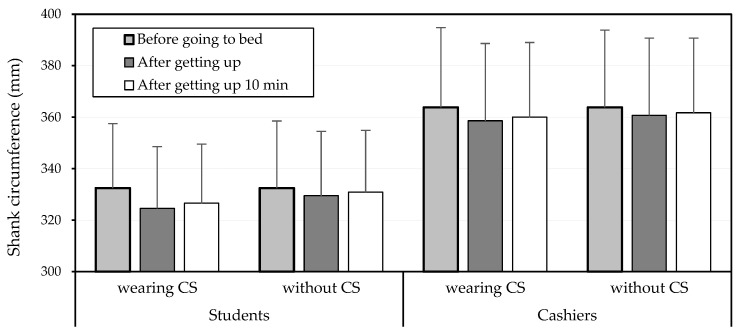
Shank circumference collected at varying time points under wearing and without compression stockings (CSs) for two participant groups.

**Figure 4 healthcare-10-01532-f004:**
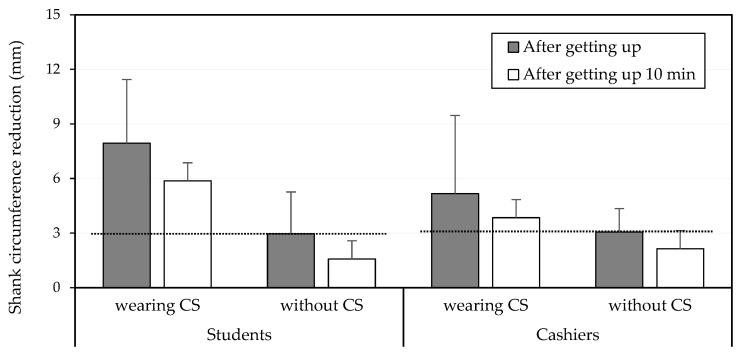
Effects of compression stockings (CSs) and measurement time on shank circumference reduction between groups. The dotted line indicates the baseline obtained after getting out of bed without CSs.

**Table 1 healthcare-10-01532-t001:** Anthropometric data of the student and cashier participants.

	Students (*N* = 10)		Cashiers (*N* = 10)			
Items	Mean	SD	Mean	SD	Difference	Independent *t* Test
Age (years)	21.9	1.5	22.7	2.5	−0.8	NS
Height (cm)	158.7	4.4	159.6	7.3	−0.9	NS
Body weight (kg)	51.5	4.1	52.9	5.9	−1.4	NS
Body mass index (BMI)	20.3	1.7	20.7	1.2	−0.4	NS
Shank circumference (mm)						
After getting up	329.5	22.4	360.7	37.6	−31.2	*t* = −2.254, *p* < 0.05
Before going to bed	332.5	25.1	363.8	38.7	−31.3	*t* = −2.146, *p* < 0.05
Difference	3.0		3.1		−0.1	NS

Notes: SD, standard deviation; NS, nonsignificance.

**Table 2 healthcare-10-01532-t002:** Main and interaction effects of shank circumference reduction obtained using three-way ANOVA.

Sources	DF	SS	MS	F	*p* Value	Power
Group (G)	1	22	22	1.96	0.166	0.281
Compression stocking (CS)	1	164	164	14.52	<0.001	0.964
Measurement time (MT)	1	63	63	5.53	<0.05	0.740
G × CS	1	38	38	3.33	0.072	0.437
G × MT	1	2	2	0.13	0.720	0.065
CS × MT	1	8	8	0.72	0.399	0.133
G × CS × MT	1	<1	<1	<0.01	0.959	0.050

**Table 3 healthcare-10-01532-t003:** Results of the discomfort (or tightness) scores and paired and independent *t* tests.

Session	Students (*N* = 10)	Cashiers (*N* = 10)	Independent *t* Test
Before going to bed	2.9 (1.2)	4.8 (2.0)	*t* = −2.576, *p* < 0.05
After getting up			
Wearing CS	2.8 (1.4)	4.4 (1.7)	*t =* −2.297, *p* < 0.05
Paired *t* test	NS	NS	
Without CS	2.0 (0.7)	3.8 (1.6)	*t* = −3.259, *p* < 0.01
Paired *t* test	*t =* 2.049, *p* < 0.05	NS	
Tightness of wearing CS	4.2 (1.7)	5.1 (1.7)	NS

Notes: Data are presented as mean (standard deviation) with units in mm; paired t tests were examined based on the data obtained before going to bed; NS, nonsignificance; CSs, compression stockings.

## Data Availability

The data presented in this study are available on request from the corresponding author. The data are not publicly available due to privacy reasons.

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
