# Peer review of "Shank Circumference Reduction by Sleep Compression Stockings in University Students and Convenience Store Cashiers"

_healthcare, 2022, doi:10.3390/healthcare10081532_

Round 1

Reviewer 1 Report

The manuscript studied whether wearing compression stockings (CS) for sleep can help to reduce occupational edema (OE). The results of the study are of practical significance and can guide people on how to wear CS. However, the following issues have to be addressed:

1.        The main conclusion of the manuscript is that wearing CS for sleep is not effective for reducing OE (Line 293). However, the main conclusion is not stated in the Abstract part of the manuscript. The authors are suggested to add the conclusion to the Abstract part.

2.        Varicose veins (VV) are closely associated with prolonged standing (Line 27) and the authors recruited 10 female university students and 10 cashiers as participants to test the CS. The reason why recruited cashiers as participants are that “Cashiering is a prime example of a job that increases vulnerability to OE and VV (Line 75)”. However, students don't have to stand for a long time every day. The authors should explain the reason for recruiting students as participants. For example, whether the reason is to set the students as a representative of the non-long-term standing crowd, to compare with the cashiers.

3.        Figure 3 shows the shank circumference (SC) reduction, the range of SC values of healthy people can be added to the Figure 3. Or the authors can explain the SC value of healthy people through words, so that it is convenient for readers to compare the SC value of volunteers with the normal SC value of healthy people.

4.        The main issue is that limited number of subjects is used, which may seriously compromise the conclusion made in the manuscript. 

Reviewer 2 Report

This study aims to investigate the effects of compression stockings (CS) during sleep in two groups (students and cashiers) on shank circumference (SC) reduction and subjective discomfort and tightness rating. The study results show that wearing CS had a significant effect on SC reduction, but the effect rapidly declined within 10 min after getting out of bed. Moreover, wearing CS may cause discomfort and tightness during sleep. To sum up, it’s an interesting and concise work with contribution. This paper is well-organized and suitable methodologies including ANOVA and the t test were used for data analysis. The content of this study is also relevant to this journal. Some comments and minor suggestions were provided before publication. 

1. Please further explain the association of SC with VV and OE in the Introduction section.

2. Please present a photo of the process of measuring SC.

3. Who measures SC? subject herself or research assistant?

4. Please provide the SC data of before going to bed in Figure 3.

5. 2.5. Statistical analysis, how to calculate SC reduction?

6. The CS used in this study has a pressure ranging from approximately 15 to 20 mmHg. The effect of sleep CS on SC might be different while using higher pressure compression stockings. It is recommended to add this description to the study limitations.

Reviewer 3 Report

Paper title “Shank Circumference Reduction by Sleep Compression Stockings in University Students and Convenience Store Cashiers” describes a simple and effective tool for alleviating varicose veins, Compression stockings which are used as a preventive measure among workers whose jobs require prolonged standing. The manuscript needs minor revision after acceptance.

1-    The abstract should contain the main conclusion

2-    Please add a clear hypothesis at the end of the introduction.

3-    The sample size in this study (10 female students and 10 female cashiers) is extremely low.

4-    Please check the number of tables and figures in the title to match those found in the text.

5-    The conclusion should be rephrased to be clearer
